# Characterisation and Effects of Different Levels of Water Stress at Different Growth Stages in Malt Barley under Water-Limited Conditions

**DOI:** 10.3390/plants11050578

**Published:** 2022-02-22

**Authors:** Zaid Adekunle Bello, Leon D. van Rensburg, Phesheya Dlamini, Cinisani M. Tfwala, Weldemichael Tesfuhuney

**Affiliations:** 1Department of Soil, Crop and Climate Sciences, University of the Free State, Bloemfontein 9300, South Africa; 21626leon@gmail.com (L.D.v.R.); cinisanitfwala@yahoo.co.uk (C.M.T.); TesfuhuneyW@ufs.ac.za (W.T.); 2Agricultural Research Council—Grain Crops, Potchefstroom 2520, South Africa; 3Department of Plant Production, Soil Science and Agricultural Engineering, University of Limpopo, Limpopo 0727, South Africa; phesheya.dlamini@ul.ac.za

**Keywords:** barley, drylands, irrigation management, plant water status, soil water status, water stress indicators

## Abstract

Malt barley is typically grown in dryland conditions in South Africa. It is an important grain after wheat, but little is known about its water requirements and, most importantly, how it responds to water stress. Determining when water stress sets in and how malt barley responds to water deficit during its growing season is crucial for improved management of crop water requirements. The objectives of this study were to evaluate the response of transpiration (T), stomatal conductance (SC), and leaf water potential (LWP) to water stress for different growth stages of malt barley and to characterise water stress to different levels (mild, moderate, and severe). This was achieved by monitoring the water stress indicators (soil- and plant based) under greenhouse conditions in well-watered and water-stressed lysimeters over two seasons. Water stress was characterised into different levels with the aid of soil water content ‘breaking points’ procedure. During the first season, at the end of tillering, flag leaf, and milk/dough growth stages, which represent severe water stress, plant available water (PAW) was below 35%, 56%, 14%, and 36%, respectively. LWP responded in accordance to depletion of soil water during the growing season, with the lowest recorded value to −5.5 MPa at the end of the milk/dough growth stage in the first season. Results also show that inducing water stress resulted in high variability of T and SC for both seasons. In the second season, plants severely stressed during the anthesis growth stage recorded the least total grains per pot (TGPP), with 29.86 g of grains. The study suggests that malt barley should be prevented from experiencing severe water stress during the anthesis and milk/dough stages for optimum malt barley production. Quantification of stress into different levels will enable the evaluation of the impact of different levels of stress on the development, growth, and yield of barley.

## 1. Introduction

One of the environmental stresses that crops are exposed to during their growth and development phases is water stress. Crop growth stages provide a sound platform to investigate how cereal crops respond to water management. As part of water management, previous studies have shown that inducing water stress during the vegetative stage of cereal enhances root development and, thus, prevents lodging in crop growth stages that occur later without substantially reducing grain yield [1]. While inducing water stress during the reproductive stage of wheat decreases yield [1,2], the degree of yield reduction depends on the duration of water stress within a specific growth stage. For instance, Zhao et al. [3] showed that mild water stress during the early vegetative growth stages and severe water stress at the maturity stage significantly affects the grain yield of wheat. Therefore, a better plant-based understanding of water stress will lead to improved water management of malt barley.

Barley (*Hordeum vulgare*) is an important grain crop after wheat in South Africa. Barley is a widely grown and highly adaptable winter cereal crop mainly used for stock feed and malt production for the brewing industry [4]. Generally, barley is grown under dryland conditions [5] because of its conservative water use, while malt barley, as a type of barley specifically, is grown mostly under irrigation. Even with its conservative water use, barley productivity is limited by severe water stress [6]. Despite the fact that the crop is an established cereal crop in the world, ranked fourth according to Baik and Ulrich [7], knowledge about its response to water stress is limited in the country. As a short-term resolution, irrigation farmers in South Africa have intuitively borrowed well-known water management principles and practices from wheat and adapted them for malt barley production. Although there are strong similarities in the agronomy of the two crops, farmers started to realise lately that there are also distinct differences, which affect their water management. One of the most important differences is the fact that the growing season of the barley cultivars is about a month shorter than that of the current wheat cultivars [8]. Thus, barley grows more rapidly than wheat through its major growth stages such as tillering, stem elongation, flag leaf, flowering, and grain ripening. This implies that the water demand of the crops will differ intrinsically over the season, as well as during its major growth stages.

One of the major limitations to crop productivity is water availability [9]. In semi-arid regions, limited water supply causes soil water deficit and this is an important plant stress factor [10]. Plant water stress under water-limited conditions influences various physiological responses including decreasing plant growth and stomatal closure, which reduce plant water use and also limit overall plant productivity [11,12,13]. Water management of field crops such as malt barley includes monitoring of plant and soil water status through the understanding of the effects of water stress on the crop. Soil water content plays a major role in crop water stress, irrigation schedule, and management. Accordingly, it is accepted that the primary cause of water stress in plants is water deficit and plants start to experience water stress either when the water supply to their roots becomes limiting or when the transpiration rate becomes intense [14]. This shows that water content determines the degree of water stress in crops, an important concept for this study. The characterisation of water stress levels of malt barley into mild, moderate, and severe is important for irrigation scheduling. It is at these critical water stress levels that the long-term term productivity of malt barley would be maintained and managed. This also involves the ability to use plant water stress indicators as tools to enhance irrigation management [15]. There is a wide variety of approaches for monitoring plant and soil water status in the soil–crop–atmospheric continuum [16]. These include soil-based measurements that determine the amount of water in the soil as well as those that are plant-based [17]. The development of water stress in the soil–crop–atmospheric continuum can be expressed through water stress indicators, viz plant-based indicators such as stomatal conductance, leaf temperature, plant–organ diameter, leaf, and osmotic water potential [18,19] as well as soil-based such as plant’s available water [16]. The accurate measurement of plant and/or soil water status is critical for studying the effects of different levels of water supply [15]. Estimating the intensity and duration of imposed water stress in the short- and long term is crucial to maximise productivity and water use efficiency of crops [20].

In South Africa, in particular, there is a lack of quantitative experimental evidence on how soil water deficit affects soil- and plant-based water stress indicators of malt barley. Therefore, this study presents a lysimeter experiment that examines the response of transpiration, stomatal conductance, and leaf water potential to different levels of water stress at different growth stages of malt barley. This is especially important in light of the growing challenge of drought in South Africa, causing water stress and resulting in the reduction in crop yields.

## 2. Results 

### 2.1. Characterisation of Water Stress (Season 1: 2011)

In the first season of the study, during the tillering stage, at the end of the mild water stress phase, *PAW* was below 35% (Figure 1). The mild water stress phase spanned up to 16 days after water was withheld. The *T* ranged between 0.32 and 0.45 mm for the mild stress phase, and from 0.45 to 0.81 mm for the moderate stress phase. The *SC* was around 60 mmol m^−2^ s^−1^ for the unwatered plant, with no regular pattern during the mild stress phase. However, due to technicalities, data of *SC* for the rest of the stage were not available. *LWP* started to decrease at −1.5 MPa, 9 days after water was withheld during the mild stress phase. The *LWP* was −1.7 MPa and −2.1 MPa at the beginning and end of the moderate stress phase, respectively. In the flag leaf stage, *PAW* was between 82 and 68% for the mild stress phase, and 68 and 62% for the moderate stress phase. *T* recorded was 1.75 mm at the end of the mild stress phase and decreased to 1.34 mm at the end of the moderate stress phase. *SC* was at 51.0, 13.9, and 34.2 mmol m^−2^ s^−1^ at the beginning of mild, moderate, and severe stress phases, respectively. *LWP* decreased from −1.6 to −2.2, −2.2 to −2.5, and −2.5 to −4.3 MPa for mild, moderate, and severe stress phases. As shown in Figure 2, *PAW* was 67 and 54% at the beginning of moderate and severe stress phases, respectively, during the anthesis growth stage. The observed ranges for mild and moderate stress phases of *T* were 0.93–0.99 mm and 0.93–1.49 mm, while those of *SC* were 25.7–60.1 mmol m^−2^ s^−1^ and 12.6–25.7 mmol m^−2^ s^−1^. The *LWP* decreased from −1.5 MPa to −2.5 MPa at the mild stress phase and further to −3.7 MPa during the moderate stress phase. At the last growth stage, milk/dough, the *PAW* was less than 51% at the end of the mild stress phase, while it was around 45% at the end of the moderate stress phase. *T* was fluctuating between 1.98 mm and 2.42 mm during the mild stress phase. However, *SC* decreased from 41.1 to 37.6 mmol m^−2^ s^−1^ during the mild stress phase and decreased further to 23.3 mmol m^−2^ s^−1^ by the end of the moderate stress phase. By the end of mild and moderate stress phases, the *LWP* was −2.0 and −3.5 MPa, respectively.

### 2.2. Characterisation of Water Stress (Season 2: 2012)

Table 1 summarises the water stress indicator values identified from the breakpoint analysis for the 2012 season before irrigating back to FC for each phase. In the tillering stage, *PAW* values decreased from 80% in the mild stress phase to 27% in the severe stress phase, which corresponded to a 53% difference between the two water stress phases. *T* values ranged between 1.26 mm in the mild phase to 1.05 mm in the severe phase. *SC* ranged from 88.08 in the mild phase to 37.15 mmol m^−2^ s^−1^ in the severe phase, while *LWP* ranged from −1.14 to −1.33 MPa in mild and severe phases, respectively. For the flag leaf stage, *PAW* values ranged between 71% in the mild phase to 58% in the moderate phase, *T* values ranged between1.59 mm (mild) to 0.87 mm (severe). *SC* ranged from 60.75 (mild) to 22.18 mmol m^−2^ s^−1^ (severe), while *LWP* ranged from −1.03 to −2.19 MPa in the mild and severe phases, respectively. For the anthesis stage, *PAW* values ranged from 73% in the mild class to 70% in the moderate class, *T* values decreased from 1.64 mm (mild) to 0.51 mm (severe). *SC* ranged from 50.98 to 9.23 mmol m^−2^ s^−1^, while *LWP* ranged from −1.99 to −3.62 MPa. For the milk/dough stage, *PAW* values ranged from 75% in the mild class to 53% in the moderate class, *T* values ranged from 2.42 mm (mild) to 0.55 mm (severe). *SC* ranged from 34.58 to 17.57 mmol m^−2^ s^−1^ while *LWP* ranged from −2.77 to −3.27 MPa. 

### 2.3. Effect of Water Stress on Grain Yield and Yield Components

The water stress treatments significantly affected the biomass, yield, and harvest index of barley for seasons 1 and 2 (Figure 3). In season 1, the effect due to water stress imposed at different growth stages on total biomass per pot (TBPP) and the total grain per pot (TGPP) was significantly different. Plants that were under water stress during the tillering stage were significantly different from those ones under water stress during the anthesis and milk/dough growth stages when it comes to TBPP. This could be associated with rapid vegetative growth during water stress. In contrast, the highest TGPP was observed in the WW treatment, which is higher than the tillering stage. The lowest grain yield was found in the water stress during the milk/dough growth stage. However, the trend of TGPP was not followed for the effect of water stress on harvest index (HI). The highest HI was recorded in WW, while the least was recorded in milk/dough treatment plants. Figure 4 also illustrates that different levels of water stress have significant effects on barley in terms of TBPP, TGPP, and HI for season 2. Irrespective of levels of water stress, the highest TBPP produced (440 g) was during the mild water stress at the anthesis stage, while the least (212.5 g) was during the severe water stress at the flag leaf stage, which is not significantly different from WW plants (215.8 g). There was no significant difference between the WW plants and plants that were mildly stressed during the anthesis growth stage in terms of TGPP. Plants severely stressed during the anthesis growth stage recorded the least TGPP (29.86 g). This scenario between the WW plants and mildly stressed plants during anthesis for TGPP was not exhibited across all the water stress levels for HI. WW plants had the highest HI of 0.55, and the plants that were severely stressed during the anthesis stage with the least HI of 0.12, which is not significantly different from the plants severely stressed during the milk/dough stage at 0.13. Seasons 1 and 2 were compared based on severe levels of water stress since this was the common level of stress found during the two seasons. Though the lowest TBPP was produced during the flag leaf growth stage, the least TGPP was produced during the anthesis growth stage during the second season. The importance of water stress during the reproductive stage, most especially during the anthesis growth stage, was emphasised with this comparison. 

## 3. Discussion

In accordance with previous studies, soil water deficit reduces *T*, *SC*, and *LWP*, of barley [21,22,23]. Determining the influence of the duration of imposed water stress is important because water deficit influences various morphological, physiological, and chemical plant characteristics [11,13,19]. An early response to water stress is crucial to aid immediate survival [24] of malt barley. The nature of plant stress responses to water-limited conditions is important since water stress resulting from withholding of water changes the physical environment for plant growth and crop physiology [15,25]. The observed response of *T*, *SC*, and *LWP* to water stress was caused by soil water being progressively depleted in the water-stressed lysimeters. The response of malt barley to water stress is dependent on the level of the water deficit in the lysimeters [13]. In the water-stressed lysimeters, water stress developed when malt barley plants absorbed less water from the surrounding environment through their roots than was transpired from their leaves [26]. As malt barley underwent its crop developmental stages, the canopy size increased concomitant with declining levels of water deficit. This led to high evaporative water demand and low water availability, causing water stress to rapidly set in leaving less time for the crop to adjust [27]. As water declined, uptake of soil water by plants was limited, causing reductions in photosynthesis, leaf area, and overall evapotranspiration component of the water budget [10]. This is because, under conditions of soil water stress, plants tend to close their stomata to prevent water losses that cannot be normally replaced from the dry soil. As a result, the rate of transpiration of the plants under water stress is reduced, and the plant becomes dependent on the available water [28]. The reduction in soil water lowers plant available water potential. This, in turn, causes a reduction in the relative water content, water potential, and turgor of cells [10]. Consequently, stomatal pores in the leaf surface progressively close, decreasing the conductance to water vapour and thus slowing transpiration and the rate at which water deficits develop [29,30,31,32]. Indeed, the ability to detect the progressive change in water status in advance of the occurrence of any stress responses is a useful characteristic of any soil- and plant-based measure of water stress [33]. Irrigation management is difficult without reliable soil- and/or plant-based measures of water stress [14].

Water stress affects crop water use or transpiration and invariably influences biomass, yield, and yield components of a crop. Yield components are important developmental mechanisms for reconstituting yield under or upon recovery from stress [34]. The stage at which a crop experiences water stress has a significant effect on the yield and yield components of that crop. In season 1, the grain yield was majorly affected during the milk/dough stage as the lowest grain yield per pot was produced during this stage. Aborted ears can be attributed to the reduction of grain yield in barley during the reproductive period [35]. Samarah [36] also reported that water stress during grain filling will significantly decrease grain yield and yield components of barley. The influence of yield components was also found in other grain crops. It is not uncommon to observe an increase in sorghum grain weight under water stress, due to a decrease in grain number per panicle, or an increase in grain number per panicle in compensation for a decrease in panicle number [37]. Clarke et al. [38] reported that water stress before seed setting in wheat affected seed numbers, while water stress after seed setting influenced the seed weight of wheat. The level of water stress is as important as the different growth stages of any crop at which water stress is experienced. Higher numbers of tillers per plant, fertile spikes, and grains per plant were observed in well-watered plants than mildly and severely stressed plants of barley [36]. However, during season 2, mostly, the effect of different levels of water stress on yield and yield components were significantly different within and different between growth stages. This suggests that identifying the growth stage, as well as the threshold of water stress within the growth stages, is very important.

## 4. Materials and Methods

### 4.1. Description of Study Site

The study was conducted at the glasshouse facility of the Department of Soil, Crop, and Climate Sciences of the University of the Free State, Bloemfontein, South Africa. This site is geographically positioned at longitude 29°02′S, latitude 26°15′E, and altitude of 1354 m. With an average annual precipitation of 547 mm and average reference evapotranspiration (ETo) of 1604 mm, the study site is classified as semi-arid [39,40]. The winters, occurring from May to August are cold and rainy summers occur between October and March is with the mean annual temperature of 15.9 °C [41]. The area has winter with the average maximum and minimum temperatures of 19 and below 0 °C. In winter, the average humidity is 54.5% during the season. 

### 4.2. Greenhouse Facility Description

The greenhouse facility had a total of 41 lysimeters made from polyvinylchloride tubes. Each lysimeter covered a ground surface area of 0.075 m^2^ (diameter = 0.31 m) with a height of 2 m. The lysimeters were filled with soil that was dug from the Kenilworth Experimental Research Station of the University of the Free State. The soil was the Plintosols [42] or Bainsvlei Amalia [43] reddish brown in colour, deep (>1.5 m), and well drained. Particle size distribution was determined using the pipette method [44], and the textural class of the soil was found to be sand, sandy loam, and sandy clay loam (Table 2). The soil was chosen because it has a good water holding capacity which plays a key role for dryland farming in this area by allowing plant growth during dry spells [45]. The A horizon of the soil was air-dried by uniformly spreading it out and turned regularly on the floor inside the glasshouse. The same procedure was employed for the B horizon. Soil drying for each of the horizons was approximately 2 weeks. The soil was then sieved (2 mm) to remove stones and clods to ensure uniform packing in all the cylinders. Known masses of dry filtered sand, A- and B-horizon soil types were packed, respectively, into cylinders. The soil was packed into the lysimeters to mimic the characteristics of the specified soil type by placing the A on top of the B and representing the approximate depths of the respective soil layers in the field. The lysimeters were first filled with filtered sand at the base to enhance drainage of excess water to drainage collectors. The average topsoil analysis indicative of the nutrient status of the soil from the field for both seasons is shown in Table 3. Each lysimeter was mounted on a 1260 Tedea–Huntleigh model load cell aluminium off-centre, loading single point with a capacity ranging between 50 and 660 kg. All the load cells were connected to an AM/16/32B relay multiplex and then connected to a CR 1000 data logger. At installation, each lysimeter was manually weighed using an electric scale. The lysimeters were repeatedly calibrated five times at the beginning of the growing season until the mass measurements satisfactorily held a coefficient of determination of more than 0.9. Extensive calibration procedures for the lysimeters in this study were reported in Bello and van Rensburg [46]. To eliminate side effects, the treatment blocks were surrounded with an additional row of black potting bags. Walkways were left between rows for easy access to the lysimeters. The lysimeters were then mulched with 0.05 m Styrofoam to minimise evaporation to negligible levels. 

During the study, the greenhouse interior climate was monitored and the temperature was set at 15 °C during the day and 5 °C at night, immediately after seeds were sown. Thereafter, the greenhouse day and night temperatures were continuously managed to mimic environmental field conditions. The day length of the interior climate was that of the outside environmental conditions since the transparent, clear glass chamber structure was completely exposed to the outside environment. The structure was also equipped with automatic vents and fans. Relative humidity, dry and wet bulb temperatures, as well as air temperature inside the greenhouse facility, were measured with a Vaisala meter. 

### 4.3. Experimental Design and Management

The experiment was conducted over a period of two cropping seasons: season 1 was from 9 June 2011 to 6 November 2011, and season 2 was from 20 May 2012 to 22 October 2012. The lysimeters were arranged in a randomised complete block design. Four replicate lysimeters were randomly assigned to each treatment for each growth stage. The barley cultivar used in this study ‘cocktail’, was obtained from the South African Barley Breeding Institute and was selected because it performs well under the dryland conditions of this area with a target seed yield of 4–5 t ha^−1^. A seed density rate of 80 kg ha^−1^ was used to plant in all the lysimeters. At planting, the lysimeter received 2:3:2 (22) N:P:K fertiliser at a rate of 90 kg N ha^−1^, 30 kg P ha^−1^ and 40 kg K ha^−1^. Topdressing with LAN at the rate of 50 kg N ha^−1^ was performed at the anthesis stage (90 days after emergence) in all treatments. Seedling emergence was observed 7 days after planting, and growth was vigorous until the initiation of the different water stress treatments. 

At the beginning of the experiment, the mass of dried soil packed in the lysimeters was recorded. The lysimeters were then saturated and covered with a 0.6 mm thick white clear plastic sheet to restrict evaporation from the lysimeter soil surface. Thereafter, the lysimeters were allowed to drain until drainage was negligible, and this was assumed at constant mass, which was observed after 48 h. Soil water that percolated to the base of the lysimeters drained through the exit pipe to a collecting bucket (D). The difference between the mass of the saturated lysimeter and the dried lysimeter was converted to volumetric water content and considered as the field capacity (FC). The plant wilting point (PWP) was estimated as the level in the drying cycle when the plants wilted. The difference between the lysimeter mass during this level and the dried lysimeter mass was used to calculate the PWP on a volumetric basis. 

Plant available water (PAW) content was then calculated as the difference between *FC* and the PWP (Equation (1)). The soil water status was reported in the form of *PAW* content.
(1)PAW=FC−PWP

Transpiration was calculated as the residual component of the water balance as follows:(2)$$T=I-\Delta KG_{final}-D$$
where *T* is transpiration, *I* is irrigation, ∆*KG _final_* represents the change in lysimeter mass, and *D* is drainage. 

### 4.4. Treatments: Levels of Water Stress Determination

The level of water stress within the different growth stages was characterised by establishing ‘breakpoints’ between periods of high and low rates of water loss, following Starr and Paltineau [47], Thompson et al. [48], and van der Westhuizen [49]. Breakpoints are important for the prediction of irrigation applications [47]. Each water stress level was identified by determining the transition stage between the high-rate water loss phase and the phase at which the rate of water loss had slowed down drastically due to the limited amount of water. The breakpoint between the high water-depletion phase and the low water-depletion phase was determined by drawing a straight line connecting water stress indicator characterised with rapid water reduction and that with slower water reduction (well watered). The intersection between these two lines was considered the breakpoint (Figure 4), following Starr and Paltineau [47]. The first and second breakpoints were then categorised into mild and moderate water stress, respectively, while the last phase was characterised as severe water stress. This procedure was performed for all the growth stages of malt barley. During the first season, the lysimeters were limited; therefore, the first season observations were used to monitor the development of the water stress using the ‘breakpoints’ procedure. The facility is equipped with a surface drip irrigation system to supply water to the lysimeters. The well-watered lysimeters were irrigated to FC every two days based on the amount of depleted soil water, which was obtained from the lysimeter load cell data. At this period, water stress was only imposed at the beginning of each of the four growth stages by withholding irrigation water until the end of the growth stages. The plants were released of stress by irrigating the lysimeters back to FC at the end of the specified growth stages. In the second season, there were enough lysimeters for each level of water stress for each growth stage. During this period, the treatments included well-watered treatment and water-stressed (mild, moderate, and severe water stress) treatments. In the water-stressed lysimeters, irrigation was stopped at the beginning of each crop growth stage and rewatered to replenish back to the FC at the end of each water stress level. This equated to the fact that the first season water stress treatments for different growth stages were equivalent to the severe water stress for the corresponding growth stage in the second season. The selected growth stages were the tillering stage, the flag leaf stage, the anthesis stage, and the milk/dough stage.

### 4.5. Plant Water Status Measurements

Plant water status was monitored through stomatal conductance (SC) and leaf water potential (LWP) of the crop after every 3 days for the duration of each growth stage and experimentation period. The SC was monitored through the use of a leaf porometer (Decagon Devices, Inc. Pullman, WA, USA). The measurements were carried out around midday between 12:00 and 14:00 h under cloudless conditions. Leaves (4) per treatment sampled for these measurements were fully expanded and fully exposed. All leaves were sampled randomly and at the same upper level on the stem of the plant. For this study, it was ensured that abaxial conductance measurement was the focus since the stomata of most leaves are on the bottom of the leaf. Before the measurement, it was ensured that the sensor head was clean and properly connected. Then, the sensor head was equilibrated to the ambient temperature during the period of the measurements. The auto mode was selected during the measurement for accurate stomatal conductance, which took only 30 s to read once the sensor head was placed on a leaf.

The LWP was measured on one mature fully expanded leaf per plant from four plants per treatment using a PMS-600 pressure chamber (PMS Instrument Company, Albany, NY, USA). During the LWP measurements, transparent plastics were used to cover the leaf before cutting to minimise water loss through transpiration. To avoid water loss from the point of incision on the leaves during the measurement of LWP, detached leaves were quickly (<30 s) mounted. The leaf was mounted on a cork, placed carefully into the chamber, and sealed with a lid. Pressure was applied slowly until water film started to appear from the point of incision, which protruded from the pressure chamber lid. A magnifying glass was used to view and determine the point in time when water appeared on the incision and the pressure reading was immediately recorded. The measurement for the plant water status was performed at midday on sunny days under cloudless conditions.

### 4.6. Yield and Yield Components

At physiological maturity, the plants were cut by hand sickle above the soil surface in each lysimeter and labelled according to the treatments per lysimeter and air-dried. The yield components such as the number of plants per pot, tillers without ear, full ear, and empty ear were determined by counting before oven drying. The total aboveground plants were oven-dried at 65 °C for 48 h and mass recorded as total biomass per pot. Ten ears per replicate for each treatment were separated, shelled, and weighed. The total ears per replicate per treatment were shelled to determine the total grain yield. The ear with sterile spikes or not developed by the crop physiological maturity were counted as an empty ear. The mass of a grain of 10 ears and grain yield was determined when the moisture content of the grain was below 15%. 

### 4.7. Statistical Analysis

Data were analysed by two-way analysis of variance (ANOVA) using SAS Program 9.4 for Windows V8 (Statistical Analysis System Institute Inc, Cary, NC, USA, 1999–2013), and differences between means were tested using Duncan Multiple range test at a probability level of 5% (*p* < 0.05). 

## 5. Conclusions

The establishment of breaking points into three classes of mild, moderate, and severe proved useful for characterising the water stress levels of malt barley. Out of all the indicators, leaf water potential responded well with soil water depletion apart from transpiration and stomatal conductance. It is recommended that this (LWP) should be used for rapid monitoring of the water stress levels. Moreover, malt barley should be prevented from experiencing severe water stress at the anthesis and milk/dough stages for optimum malt barley production. Quantification of stress into different levels will enable the evaluation of the impact of different levels of stress on the development, growth, and yield of barley. This will provide the opportunity to identify the level of water stress or threshold that is allowable before the next irrigation without compromising the production. Describing the occurrence and consequence of water stress in malt barley is especially important for farmers in South Africa, as it is typically grown under dryland conditions. A better plant-based understanding of water stress will lead to improved water management of malt barley.

## Figures and Tables

**Figure 1 plants-11-00578-f001:**
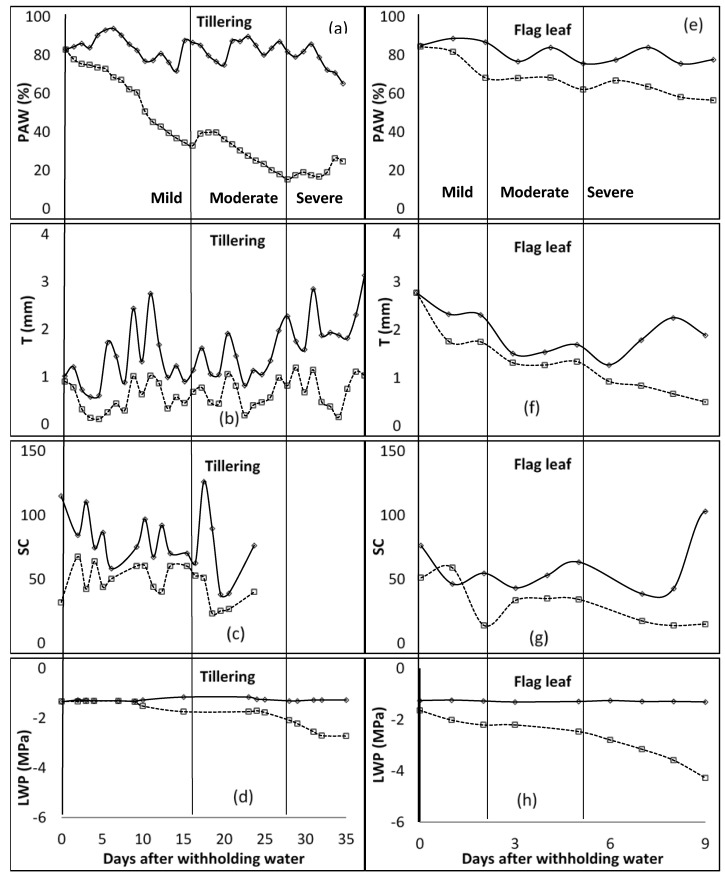
Distribution of plant available water content (*PAW* [(**a**)—tillering; (**e**)—Flag leaf]), transpiration (*T* [(**b**)—tillering; (**f**)—Flag leaf]), stomatal conductance (*SC* [(**c**)—tillering; (**g**)—Flag leaf] (mmol m^−2^ s^−1^)), and leaf water potential (*LWP* [(**d**)—tillering; (**h**)—Flag leaf]) for tillering and flag leaf growth stages of barley. The vertical line characterises the water stress levels. These lines signify the end and or the beginning of a stress level (mild, moderate, and severe—specified for each column). WW—well watered, solid lines; stressed—broken lines.

**Figure 2 plants-11-00578-f002:**
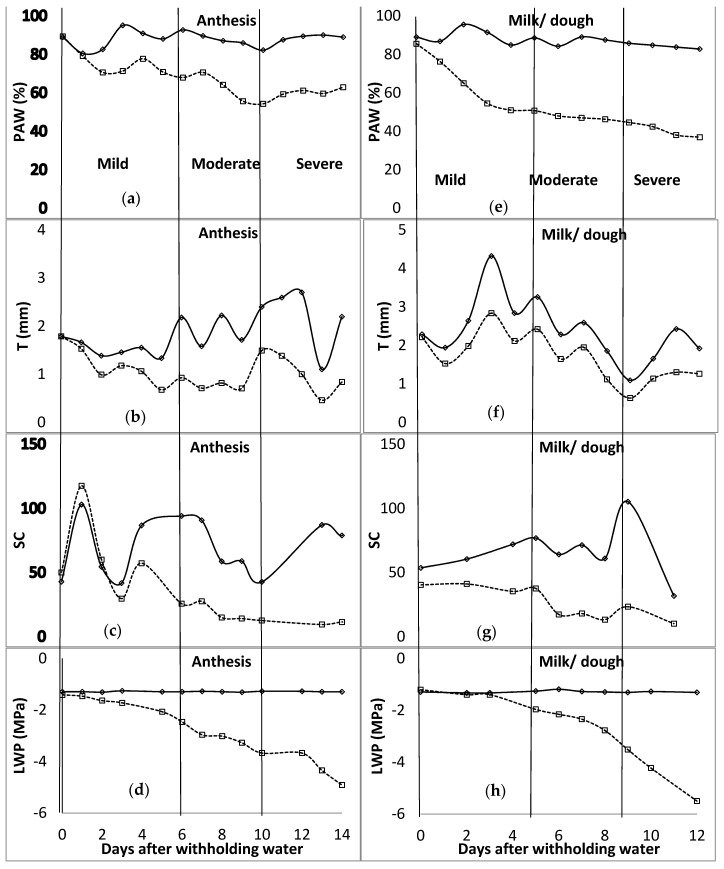
Distribution of plant available water content (*PAW* [(**a**)—anthesis; (**e**)—milk/dough]), transpiration (*T* [(**b**)—anthesis; (**f**)—milk/dough]), stomatal conductance (*SC* [(**c**)—anthesis; (**g**)—milk/dough] (mmol m^−2^ s^−1^)), and leaf water potential (*LWP* [(**d**)—anthesis; (**h**)—milk/dough]) for anthesis and milk/dough growth stages of barley. The vertical line characterises the water stress levels. These lines signify the end and or the beginning of a stress level (mild, moderate, and severe—specified for each column). WW—well watered, solid lines; stressed—broken lines.

**Figure 3 plants-11-00578-f003:**
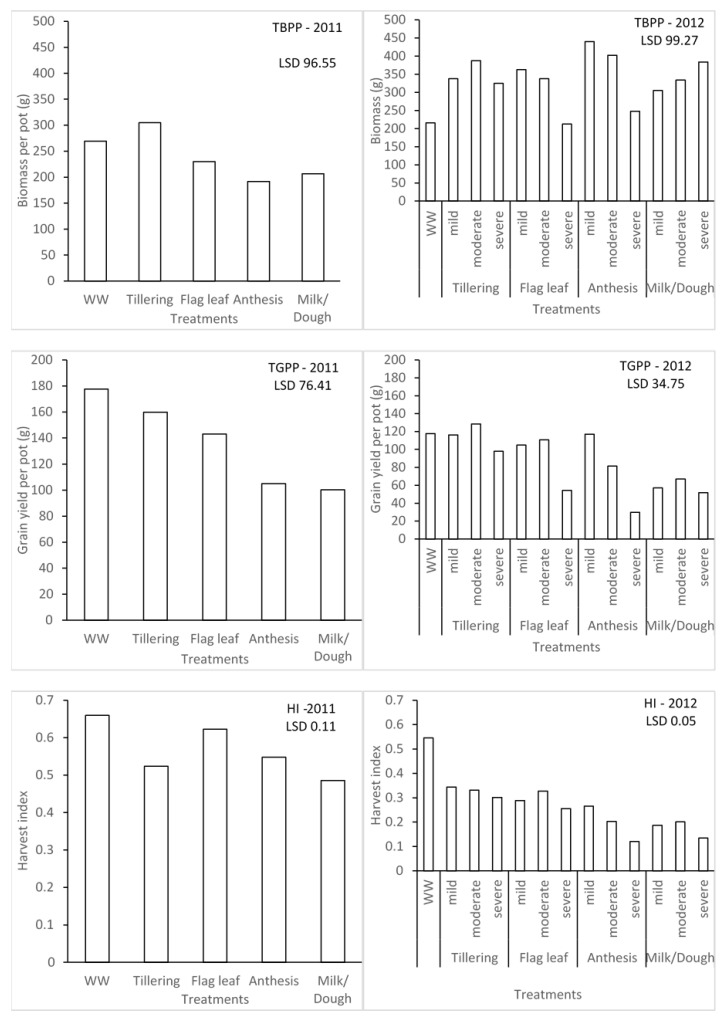
Effect of different water stress levels on biomass production, grain yields, and harvest index of malt barley.

**Figure 4 plants-11-00578-f004:**
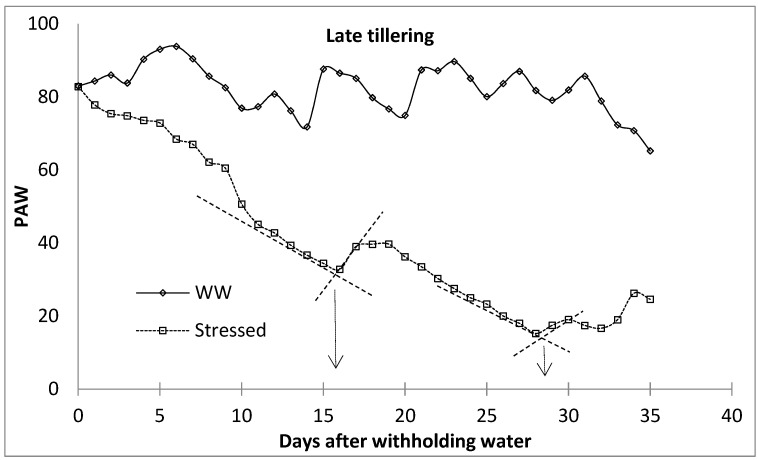
Breaking point procedure to identify water stress level phases (following Starr and Paltineau [47]. WW—well watered.

**Table 1 plants-11-00578-t001:** Characterisation of water stress values of plant available water content (*PAW*), transpiration (*T*), stomatal conductance (*SC*), and leaf water potential (*LWP*) at the end of each stress phase for the four growth stages (tillering, flag leaf, anthesis, and milk/dough) of barley.

Water Stress Indicator	Water Stress Level	Tillering	Flag Leaf	Anthesis	Milk/Dough
*PAW* (%)	Mild	80.12	70.63	72.86	75.44
	Moderate	43.83	57.51	70.9	53.28
	Severe	27.53	46.75	64.32	49.55
*T* (mm)	Mild	1.26	1.59	1.64	2.42
	Moderate	1.15	1.45	1.53	1.76
	Severe	1.05	0.87	0.51	0.55
*SC* (mmol m^−2^ s^−1^)	Mild	88.08	60.75	50.98	34.58
	Moderate	83.45	45.21	31.15	24.53
	Severe	37.15	22.18	9.23	17.57
*LWP* (MPa)	Mild	−1.14	−1.03	−1.99	−2.77
	Moderate	−1.23	−1.26	−2.5	−2.93
	Severe	−1.33	−2.19	−3.62	−3.27

**Table 2 plants-11-00578-t002:** Average physical properties of the packed soil for different depths in the lysimeters.

Soil Depth (mm)	Textural Class	Total Sand	Clay	Silt	Bulk Density	Porosity
		(%)	(%)	(%)	(g cm^−3^)	(%)
0–300	Sand	90	8	2	1.64	32
300–600	Sandy loam	82	14	4	1.52	46
600–900	Sandy loam	82	14	4	1.52	46
900–1200	Sandy loam	82	14	4	1.52	46
1200–1500	Sandy clay loam	76	20	4	1.57	48
1500–1800	Sandy clay loam	76	20	4	1.57	48

**Table 3 plants-11-00578-t003:** Average topsoil analysis indicative of nutrient status of the specified soil type in the lysimeters prior to cropping of season 1 (2011) and season 2 (2012).

Season	N	P	K	C	Ca	Mg	Na	Zn	pH (H_2_O)
	----------------------------------mg kg^–1^-----------------------------------								
Season 1	396.38	26.264	184.8	1328	827.6	230.8	34	5.36	7.48
Season 2	332.3	26.39	154	1070	604	158	22	4.7	7

## Data Availability

Data is contained within the article.

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
