# Peer review of "Characterisation and Effects of Different Levels of Water Stress at Different Growth Stages in Malt Barley under Water-Limited Conditions"

_plants, 2022, doi:10.3390/plants11050578_

Round 1

Reviewer 1 Report

Researchers made this kind of study during the last decade, but this manuscript lacked some methodologic criteria and scientific innovation.

Title: Characterization and effects of different levels of water stress at different growth stages in malt barley under water-limited conditions

Abstract

The abstract is not sufficiently informative.

Obs. The abstract must consist of a brief introduction of the theme, objective, methodology, result, and conclusion;

Key-words

I believe that one should not repeat words from the title.

Introduction

References used to justify this research do not support the need for this investigation. An analysis of the environmental factors that affect stomatal conductance and plant water status is absent. Stomatal conductance increased with rising temperature despite decreasing leaf water potential, increasing plant transpiration and CO2 concentration.

Theoretical considerations about stomatal conductance are needed.

Material and Methods

2.2 Greenhouse facility description

Soil physical characteristics are necessary.

2.5 Plant water status measurements

Which level of CO2 concentration was used?

Plant measurements, such as LAI and plant height, are necessary.

How were the treatments characterized? Mild, moderate, and severe?

Results

Graphics and tables are confused and lacking in information. A relation between stomatal conductance and VPD is necessary.

How does the soil water content influence stomatal conductance?

Author Response

Good day,

Thanks for your creative and thoughtful comments. The comments has been attended to and attached to this mail. 

Regards, 

Reviewer 2 Report

  1. Materials and methods

2.1. Description of study site

how is the average humidity in this area? And max and min temperatures?

2.5. Plant water status measurements

  • Leaves (4) per treatment samples for these measurements were fully expanded and fully exposed.: I understand that you measured on 4 leaves, but and in how many plants?
  • Figure 1. you have a different size of letters in the foot
  • The LWP was measured on one mature, how many plant did you measure?
  • Figure 1: WW= well-watered? WW means, not found in text

  1. Results

3.1. Characterization of water stress (Season 1: 2011)

  • The mild water stress phase spanned up to 16 after water was withheld. Day 16??? I suggest write “day” for easier reading
  • figure foot 2 and 3: you should be specified what each line is (continuous and discontinuous).

  1. Discussion

  • the lines 326 to 336 is for introduction not for discussion.
  • should be compared more with other similar studies

Author Response

(The authors gave the same response as above.)

Reviewer 3 Report

I have the opportunity to evaluate your manuscript. It is well written, the topic is very interesting, the experimental design is very well conducted, results are clearly described and conclusion answers the objectives.

Thus I recommend your manuscript for publication. Keep on your good work.

Author Response

Good day,

Thanks for your encouraging comments. 

Regards, 
